# Trait Energy and Fatigue Modify the Effects of Caffeine on Mood, Cognitive and Fine-Motor Task Performance: A Post-Hoc Study

**DOI:** 10.3390/nu13020412

**Published:** 2021-01-28

**Authors:** Daniel T. Fuller, Matthew Lee Smith, Ali Boolani

**Affiliations:** 1Department of Mathematics, Clarkson University, Potsdam, NY 13699, USA; fullerdt@clarkson.edu; 2Center for Population Health and Aging, Texas A&M University, College Station, TX 77843, USA; matthew.smith@tamu.edu; 3Department of Environmental and Occupational Health, School of Public Health, Texas A&M University, College Station, TX 77843, USA; 4Department of Physical Therapy, Clarkson University, Potsdam, NY 13699, USA; 5Department of Biology, Clarkson University, Potsdam, NY 13699, USA

**Keywords:** trait energy, trait fatigue, caffeine, moods, cognitive tasks, psychomotor tasks

## Abstract

Multiple studies suggest that genetic polymorphisms influence the neurocognitive effects of caffeine. Using data collected from a double-blinded, within-participants, randomized, cross-over design, this study examined the effects of trait (long-standing pre-disposition) mental and physical energy and fatigue to changes in moods (Profile of Mood Survey-Short Form (POMS-SF), state mental and physical energy and fatigue survey), cognitive (serial subtractions of 3 (SS3) and 7 (SS7)), and fine-motor task (nine-hole peg test) performance after consuming a caffeinated beverage and a non-caffeinated placebo. Results indicate that trait mental and physical fatigue and mental energy modified the effects of caffeine on vigor, tension-anxiety, physical, and mental fatigue. Additionally, we report that those who were high trait physical and mental fatigue and low-trait mental energy reported the greatest benefit of caffeine on the SS3 and SS7, while those who were high trait mental and physical fatigue reported the greatest benefit of consuming caffeine on fine-motor task performance. The results of our study suggest that trait mental and physical fatigue and mental energy modify the acute effects of caffeine among a group of healthy, young adults and should be measured and controlled for by researchers who choose to study the effects of caffeine on acute moods and cognitive and fine-motor task performance.

## 1. Introduction

Fatigue is a common, costly, and poorly understood problem, which affects approximately 45% of the United States (US) population [1]. It has been estimated that fatigue costs employers over $136 billion per year in lost productivity [2]; however, these estimates do not account for fatigue-related driving and other accidents [3,4], poor medical performance [5], and negative health outcome [6]. Fatigue is also underreported in medical care [7] and has been linked to many diseases and disorders [8]. Despite the high financial and social costs of fatigue, it is a poorly understood problem. For example, until recently, most researchers viewed energy and fatigue on a bipolar continuum (e.g., if an individual is not energetic, then they are fatigued). However, Loy and colleagues [9] recently provided evidence that energy and fatigue are two distinct moods (e.g., an individual can be energetic and fatigued simultaneously), with multiple studies since showing that feelings of energy and fatigue are distinct yet overlapping constructs [10,11,12], with their own mental and physical components [13,14]. Although we are aware of multiple interventions, such as exercise [15], caffeine [16,17], and sleep [18], that increase feelings of energy and/or decrease feelings of fatigue, evidence regarding the effectiveness of these interventions is mixed. Additional research is needed to better understand the inter- and intra-individual differences in the efficacy of these interventions.

One common acute intervention for feelings of low energy and high fatigue is caffeine [16,17]. Caffeine is consumed in various forms, with coffee and energy beverages being the most prevalent. However, to our knowledge, the release and absorption rates of caffeine from coffee and energy beverages is the same [19]. Multiple studies have reported that there are inter-individual differences in the influence of caffeine on moods [20,21,22,23,24]. These studies [20,21,22,23,24] have reported that ADORA2A gene polymorphism may be primarily responsible for the inter-individual variations in the effects of caffeine on anxiety [21,24] and physical fatigue [22,23]. While substantial evidence exists on the role of the ADORA2A gene in determining the ergogenic effects of caffeine [20,21,22,23,24], identifying gene mutation prior to administering caffeine as an ergogenic aid may be impractical for most nutrition researchers and practitioners. Therefore, finding low-cost validated methods to identify factors that influence the ergogenic effect of caffeine is pragmatic and desirable to both practitioners and researchers. 

One such potential measure, is trait (long-standing pre-disposition) mental and physical energy and fatigue, a construct only recently reported in the literature [13,14]. While the authors of this study are aware of only one study that examines the effects of trait mental and physical energy and fatigue on moods [14], that study reports that trait mental and physical energy and fatigue moderates the effects of sleep on state energy and fatigue [14]. These findings raise an interesting question about whether trait mental and physical energy and fatigue may also modify the intensity or effects of other interventions (e.g., caffeine) on changes in self-reported mood and/or objective measures of mental energy during the performance of a common mental test battery [25,26,27,28,29] used in nutritional science research. Based on this premise, we re-examined data from a previously published study [28] that investigated whether an adaptogenic-rich caffeine-containing beverage would modulate the effects of caffeine on self-reported mood and objective measures of energy and fatigue.

Therefore, the aims of our current analyses were to determine whether trait mental and physical energy and fatigue influence the rate of change in (1) state moods, (2) cognitive task performance (objective mental energy measure), and (3) fine motor task performance, when participants perform a commonly used mental energy test battery [25,26,27,28,29] on days when they consumed caffeine compared to the days when they consumed placebo. To address these aims, we performed a post-hoc analysis of our previous study [28] limited to the days when participants consumed placebo and the active comparator (caffeinated) conditions only.

## 2. Methodology

### 2.1. Study Design and Study Products

A full description of the methodology has been previously published [28]. In our previous double-blinded, placebo-controlled, within-participants, randomized cross-over study, we examined the effects of three 60 mL interventions: (1) a placebo; (2) an active comparator (caffeine); and (3) e+^TM^ shot (e+ shot, Isagenix International, LLC, Gilbert, AZ, USA). For the purposes of this study, we only examined the placebo and active comparator (caffeine). To ensure effective blinding in the main study, none of the scientists conducting the study or analyzing the data were aware of treatment assignments. All treatments were delivered in identical unmarked white containers with a black top. The placebo and caffeine beverages had the same base components as e+ shot (purified water, apple juice concentrate, glycerin, pomegranate juice concentrate, natural flavors, malic acid, potassium sorbate, and sodium citrate) to which either 0 mg (placebo) or approximately 98 mg synthetic caffeine (caffeine) were added. Quantitation of caffeine in the study products was verified according to Eurofins Scientific Inc. (Des Moines, IA, USA).

### 2.2. Screening and Participants

After receiving Clarkson University Institutional Review Board (IRB) approval (approval # 16-34.1), participants were recruited from 16 September 2015 to 9 August 2016 from a small private university and local community using in-class announcements, bulletin boards, electronic listservs, flyers at local businesses, and word of mouth. Potential participants were invited to complete a screening questionnaire administered online using SurveyMonkey Inc. (San Mateo, CA, USA).

The exclusion criteria were as follows: under the age of 18 or over the age of 45; self-reported body mass index (BMI) > 30; above-average feelings of energy on the Profile of Mood Survey- Short Form (POMS-SF) (scores > 12); high caffeine consumers (>21 servings of 170.5–341 mL caffeine beverages per week; high consumption of polyphenols (>100 total combined servings per week); self-reported chronic physical or mental health condition requiring prescription or over-the-counter medication (excluding contraception) on a continual basis; pregnant or reported a chance of being pregnant; allergy to caffeine; current smoker; or consumption of nutritional supplements (i.e., herbs, vitamins, or creatine, not including supplementation of protein without caffeine).

Volunteers not excluded by the screening were invited to the testing facility. All participants read and signed the informed consent form. Participants were informed that they would be taking part in a study investigating the effects of caffeine beverages on mental function, blood pressure, heart rate and fine motor control. Thirty (17 women and 13 men) participants completed the original study and were all included in our current post-hoc analysis. Characteristics from the sample (*n* = 30) are reported in Table 1.

The average reported nightly sleep during the month prior to the study was 7.6 ± 0.8 h. The number of hours of reported sleep the night before each of the testing sessions did not significantly differ from between conditions (*t* = 0.38, *p* = 0.71); placebo (6.5 ± 1.3 h); caffeine (6.4 ± 1.1 h). Participants appeared to be low consumers of caffeine (4.2 ± 3.8 servings per week) and polyphenols (56.74 ± 8.16 servings per month). All participants were asked to refrain from consuming caffeine 24 h prior to testing, and salivary analyses were completed on each testing day to confirm compliance.

### 2.3. Measures

During the survey to determine eligibility for the study, participants completed a series of assessments. For the purposes of this study, we chose to analyze the influence of trait mental and physical energy and fatigue on the testing day measures. On testing days, participants completed a series of mental energy tests consisting of self-reported motivation and mood measures and computerized cognitive tasks of sustained attention. Additionally, we measured fine motor task performance using the nine-hole peg test. 

All testing was performed in a seated position in a thermoneutral (72 ± 0.8 °F/22.2 ± 0.8 °C), private lab setting, with sound attenuation and controlled lighting. Visual stimuli were presented that required a finger response. Participants used the keyboard to respond to information presented on a 17′′ screen on an Alienware laptop (17 R2 Model #P43F, Roundrock, TX, USA). Prior to each cognitive task, participants were given on-screen instructions about how to perform the task and asked to press the “enter” key if they understood the directions or to get help from a researcher if they were uncertain. All cognitive tests were performed using the Membrain Platform (PsychTechSolutions, Potsdam, NY, USA) using Java-coded software. Results from the cognitive tasks were downloaded into Microsoft Excel and two research assistants independently manually re-arranged data for analysis.

For the purposes of this study, we only describe the measures used in the present analyses. For a full list of pre-testing and testing day measures, the reader may refer to Boolani et al. [13] and Boolani et al. [28], respectively.

#### 2.3.1. Pre-Testing Measure

Trait Mental and physical energy and fatigue: The trait aspect of the mental and physical state and trait energy and fatigue scale was used to collect pre-testing information on trait (long-standing pre-disposition) mental and physical energy and fatigue. The trait component, which references how the respondent usually feels, contains 12 total items with three items for each of the four trait outcomes (physical and mental energy and fatigue). Representative statements include: “I feel I have energy” and “I have feelings of being worn out.” Responses were collected on a 5-point scale ranging from “never” to “always”. In other studies, the Cronbach’s alpha coefficients range from 0.82 to 0.93 [13,14,30]. With the current data, alpha coefficients ranged from 0.73 to 0.88 (trait mental energy = 0.73, trait mental fatigue = 0.88, trait physical energy = 0.75, trait physical fatigue = 0.84).

#### 2.3.2. Testing Day Measures

(1) State Moods: The 30-item Profile of Mood Survey–Short Form (POMS-SF) was used to assess mood states in the moment using a five-point scale ranging from “Not at all” (scored as 0) to “Extremely” (scored as 4). Scores from these questions were used to calculate the different components of tension/anxiety (α = 0.364), depression (α = 0.598), anger (α = 0.519), vigor (α = 0.922), fatigue (α = 0.863), and confusion (α = 0.419). All dimensions were made of five items (i.e., tension= tension + shaky + uneasy + nervous + anxious) [31]. 

(2) State Mental and Physical Energy and Fatigue: The state aspect of the mental and physical state and trait energy and fatigue scale was used to measure feelings in the immediate moment [29]. Like the trait component, the state component had the same 12 items as the trait scale, but this time a 0 to 100 Visual Analog Scale (VAS). However, due to limitation in data collection techniques, the scale was modified to a 0–10 Likert Scale anchored by “absence of feelings (left end, scored as 0) and the “strongest intensity of feelings” (right end, scored as 10). The modification is the same as Boolani et al. previously used (10,29). The Cronbach’s alpha for this current study was between 0.707 and 0.874 (state physical energy = 0.785, state physical fatigue= 0.837, state mental energy = 0.707, and state mental fatigue = 0.874). 

(3) Serial Three and Serial Seven subtraction tasks: Participants were asked to silently subtract backwards in three’s or seven’s from a random starting number between 800 and 999 that was presented on the computer screen (Tahoma Regular font, size 20 pt). Participants were instructed to type their answers as quickly and as accurately as possible. After each answer was entered by the participant, the number was cleared from the screen. Participants were allowed to complete as many attempts as possible in two minutes [26,27]. For the purposes of this study, we did not analyze the number of correct responses because all tests had >97.5% response accuracy. Therefore, we only analyzed the total number of attempts for these serial subtraction tests. The maintained correct response rate suggests that participants sacrificed speed for accuracy as they became more fatigued.

(4) Fine motor control: The validated nine-hole peg test of finger dexterity was used to measure fine motor control [32]. The 12 × 12 cm wooden pegboard contained nine holes and was placed on the desktop in front of the seated participant. There were nine cylindrical pegs that were placed on the desktop outside of the container on the right or left side of the board for when the participant’s right- and left-hand dexterity was tested. Participants were instructed to place one peg into the pegboard holes one at a time and then remove each peg one at a time, as fast as possible. The first test was performed with the dominant hand and the next with their non-dominant hand. Each test was performed twice. For the purposes of this study, we examined the mean scores from the two tests. Results are presented as the average time (measured in seconds) of the two times for the dominant hand (DH) and the average of the results of the two times for the non-dominant hand (NDH).

### 2.4. Procedure

The study consisted of a familiarization day (~1 h) followed by three testing days (~3 h) scheduled a minimum of 48 h apart. Each day started between 6 and 8 am and was scheduled within ±30 min of the familiarization day to account for diurnal variations [33]. Participants were advised to get their typical amount of sleep and asked to refrain from caffeine and alcohol consumption the night before testing. Additionally, they were asked to refrain caffeine and alcohol consumption the day of testing.

Familiarization Day: To reduce the experimental error that may occur due to learning effect, participants were asked to come to lab for a practice session where they completed a single trial run of all the daily assessments. These data were not included in any of our analyses. On familiarization day, we also measured participants’ characteristics. Height was measured using a stadiometer and weight was measured using a digital scale (Tanita TBF-410, Tanita Corporation, Tokyo, Japan).

Testing days 1–3: Using randomizer.org, participants were randomly assigned to the order in which the beverage was administered. Participants came to lab each testing day and completed a series of surveys that asked them about their previous night’s sleep, food, beverage, and drug consumption over the prior 24 h. Participants who reported ±2 h of their usual sleep duration (reported during the screening) were not tested that day and rescheduled. Those who reported drug use or the consumption of caffeine-containing foods or beverages the night before were also rescheduled. After screening, participants were instructed to accumulate ~2 mL of saliva in a 10 mL test tube. Baseline measures of sustained attention (cognitive task measures as part of the mental energy test battery), motivation, mood, blood pressure (BP), heart rate (HR), and fine motor task performance were obtained (Figure 1). After baseline testing, participants were administered one of the beverages and instructed to consume it within 2 min. Following the administration of the beverage, participants were given a 28 min break and were not allowed to participate in strenuous physical or mental activity or consume additional snacks or beverages. Three additional 27 min mental energy battery tests were completed, with 10 min rest breaks between each test battery (Figure 2). At the end of the last test battery, participants provided a post-test saliva sample using the same drool-down method described above.

### 2.5. Data Treatment and Statistics

#### 2.5.1. Data Handling

The medians for each trait variable (Trait Physical Energy = 7.0, Trait Physical Fatigue = 3.5, Trait Mental Energy = 6.0, Trait Mental Fatigue = 3.0) were identified, and then surrogate variables were created to represent these in dichotomous format with their values at 0 up to the 50th percentile and 1 otherwise. All data processing and analyses were done with R version 4.0.0. Scripts for these tasks can be found at Trait-Caffeine study.

#### 2.5.2. Primary Analysis

Linear mixed-effects regressions were performed on each set of variables to test for significant differences across testing groups before and after they consumed the intervention [34,35]. For one-way effects, this test provides equivalent results to a type III repeated-measures ANOVA, computing and comparing the estimated population marginal means for two time points. For two-way effects, singularly significant mixed-effects results are interpreted in a straightforward manner, unlike those of an ANOVA, which are interpreted as crossover effects. All tests and relevant marginal means figures were computed and generated using the lmerTest package in R. Statistically significant relationships are presented in manuscript tables; however, a table containing values for all statistical relationships can be found in the Appendix A (Appendix A).

## 3. Results

After screening 1035 surveys for eligibility, 43 participants qualified for the study and were randomly allocated to the order in which they would receive the intervention. Due to logistical issues, 13 participants started the study but did not complete it. A total of 30 participants completed both days of the study, and their data were used for our final analysis (Figure 3). Recruitment and data collection lasted from 25 September 2015 to 10 December 2016 until 30 participants had completed both days of treatment. There were no harms or unintended consequences for any of the interventions. No participants pre-testing salivary caffeine levels >0.05 µg/mL, suggesting that all participants followed instructions of abstaining from caffeine prior to testing day. Post-hoc power analysis revealed a calculated power >0.90 for all analyses.

### 3.1. Trait Influence on State Mood

When testing the caffeinated and placebo conditions together, we report that independent of caffeine, high trait physical energy increased fatigue (β = 1.70, 95% CI: 0.43, 3.00) and confusion (β = 0.830, 95% CI: 0.033, 1.628) and decreased POMS vigor (β = −3.20, 95% CI: −5.43, −1.58) and motivation to perform mental tasks (β = −1.469, 95% CI: −2.827, −0.111). Low trait physical energy increased state physical fatigue (β = −4.90, 95% CI: −5.724, −1.33) and state metal fatigue (β = −4.50, 95% CI: −5.67, −1.33) with performance of the mental test battery (Table 2). 

Independently, caffeine was found to increase POMS vigor (β = 1.30, 95% CI: −0.82, 1.95), POMS tension/anxiety (β = 0.63, 95% CI: −0.07,0.54), and POMS anger (β = 0.400, 95% CI: −0.013, 0.787). However, low trait physical fatigue and caffeine increased POMS tension/anxiety scores (β = −0.80, 95% CI: −0.23, 0.63), meaning that low trait physical fatigue amplifies the effects of caffeine on POMS tension/anxiety (Figure 4). The interaction between all the trait moods and caffeine were not significant for vigor and anger, suggesting that trait did not modify the effect of caffeine on vigor and anger (Table 2). 

Low trait mental energy and caffeine together decreased state physical fatigue (β = 5.40, 95% CI: −0.23, 6.35) and mental fatigue (β = 5.20, 95% CI: −0.27, 6.27), while alone neither had any statistically significant independent effect. We also found that low trait mental fatigue and caffeine increased feelings of depression (β = −0.339, 95% CI: −0.632, −0.047) and motivation to perform mental tasks (β = −1.312, 95% CI: −2.365, −0.260) (Table 2).

### 3.2. Trait Influence on Cognitive Task Performance

When testing the caffeinated condition and placebo together, we found that while trait physical fatigue did not independently influence attempts of serial subtraction 3, the interaction between caffeine and trait physical fatigue was positive (β = 4.40, 95% CI: −3.23, 6.50), suggesting that those who reported high trait physical fatigue reported a greater benefit of consuming caffeine. While high trait mental energy increased serial subtraction 3 attempts (β = 9.20, 95% CI: −0.07, 12.27), the interaction between trait mental energy and caffeine was negative (β = −6.20, 95% CI: −1.97, 10.79). This suggests that those with low trait mental energy enjoyed greater benefits of caffeine on serial subtraction 3 attempts. While consuming caffeine reduced subtract 3 attempts (β = −2.9, 95% CI: −1.16, 4.80) independently of trait mental fatigue, caffeine and high trait mental fatigue together increased subtract 3 attempts (β = 5.70, 95% CI: −3.42, 6.66). This suggests that participants with high trait mental fatigue received the most benefit from consuming caffeine on the serial subtract 3 (Table 2). 

### 3.3. Trait Influence on Fine Motor Task Performance

High trait physical fatigue and caffeine together decreased dominant hand average completion time by 9.4 s (β = −9.40, 95% CI: −8.05, 9.25) and nondominant hand average completion time by 8 s (β = −8.00, 95% CI: −13.27, 4.14). High trait mental fatigue and caffeine together decreased dominant hand average completion time by 8.9 s (β = −8.90, 95% CI: −12.11, 5.82) and nondominant hand average completion time by 8.0 s (β = −8.00, 95% CI: −17.75, −0.08) (Figure 5). These results imply that those with high trait mental and physical fatigue reported the most benefit of consuming caffeine on the 9-hole peg test (Table 2).

### 3.4. Post-Hoc Salivary Analysis

There were no statistically significant effects of Trait Physical Energy (*p* = 0.487), Trait Physical Fatigue (*p* = 0.312), Trait Mental Energy (*p* = 0.458), or Trait Mental Fatigue (*p* = 0.837) on pre-post salivary caffeine composition.

## 4. Discussion

To our knowledge, this post-hoc analysis is the first study to analyze the influence of trait mental and physical energy and fatigue on the neurocognitive effects of caffeine during the performance of a commonly used mental energy test battery. The results of our study provide new evidence suggesting that trait energy and fatigue may explain the interpersonal differences in the effects of caffeine on mood, cognitive and fine-motor task performance.

### 4.1. Trait Influence on State Moods

The results of our study indicate that trait physical energy by itself influences mood responses as a result of performing the mental test battery. For example, we found that those who reported being high trait physical energy reported a greater decline in POMS vigor (energy) scores and a greater increase in POMS fatigue and confusion scores and motivation to perform mental tasks. These results suggest that participants who report being normally very physically energetic may respond more negatively to repeatedly performing this mental task battery. However, in our study, when we split fatigue into mental and physical aspects of fatigue, those with low trait physical energy reported greater declines in both physical and mental fatigue during the repeated performance of this mental test battery. This relationship is contrary to their responses to the POMS fatigue scores. While we cannot explain these differences, it raises questions as to the constructs measured by the two surveys. While the POMS-SF and the state mental and physical energy and fatigue surveys are widely used together [16,17,23,28,29], in our study, their results were incongruent. With high alphas (>0.84), the consistency of the item responses within our study is strong, which suggests responses were not biased by measurement error.

As expected, when comparing the caffeine to placebo condition, we found that caffeine increased feelings of vigor [16,17,29], anger [29,36], and anxiety [24,29,37]; however, our findings were unique in that we reported that trait physical fatigue moderated the effects of caffeine on feelings of anxiety. Our findings suggest that those who were low trait physical fatigue reported an increase in feelings of anxiety on days they consumed caffeine compared to days they consumed placebo, as shown in Figure 4. The opposite is true for those who reported high trait physical fatigue, in that these individuals reported reduced feelings of anxiety on days they consumed caffeine compared to days they consumed the placebo beverage. Our findings may explain the results of the study by Childs and colleagues [24], where most, but not all, participants reported increased feelings of anxiety after a high dose of caffeine. Another associated interaction in our results was the fact that individuals with low trait mental energy reported greater decreases in feelings of mental and physical fatigue after the consumption of caffeine. Our results suggest that those who normally report feeling mentally energetic may not receive the same anti-fatiguing benefits of consuming caffeine as those who normally report not feeling mentally energetic. These two findings may be related because we find that the ADORA2A gene, a dopaminergic gene, is associated with both feelings of energy [9] and anxiety [24]. Therefore, we may hypothesize that genetic polymorphisms associated with the fatigue may modify the expression of the ADORA2A gene with caffeine consumption. 

We also reported that those who were low trait mental fatigue reported increased feelings of depression after consuming caffeine. These results may explain why some of the participants in a study by Dawkins and colleagues [38] reported an increase in feelings of depression after consuming caffeine. Additionally, we found that those who were low trait mental fatigue reported greater increases in motivation to perform mental tasks, suggesting that those who are normally mentally fatigued may receive a boost in motivation compared to those who are not normally mentally fatigued. The results of our study might explain why cyclists who reported feeling mentally fatigued reported increases in motivation to perform mental tasks after consuming caffeine, while those who were not mentally fatigued at baseline reported no increases in motivation [39]. Other studies that have examined their population as a whole and have not split them up by those who normally feel mentally fatigued compared to those who do not, have reported inconsistent findings for the effects of caffeine on motivation [29,40], including the primary study from this data [28]. All of the aforementioned studies [28,29,40] used very similar protocols and similar amounts of caffeine (66–100 mg), yet they had differing results as it relates to the effects of caffeine on motivation to perform mental tasks. Our results suggest that studies should control for trait mental and physical energy and fatigue as those traits may modify the effects of caffeine on motivation. 

### 4.2. Trait Influence on Cognitive Task Performance

Our results found that overall, those who were high trait physical and mental fatigue or low trait mental energy enjoyed the benefits of caffeine consumption on the serial subtraction 3 task, while caffeine only had an effect on participants who were high trait mental fatigue on the serial subtraction 7 task. These results were intuitive in that those who need the biggest boost to perform cognitive tasks received it when they consumed caffeine. Our results were also in line with previous findings that caffeine by itself increased serial subtraction attempts [29]. An interesting finding from our study was that those who were normally mentally energetic reported improvement in serial subtraction 7 attempts during this protocol even without the benefit of consuming caffeine. Although the protocol in our study has been validated [25] and used in multiple previous studies [16,17,26,27,29], our findings suggest that not everyone may report the intended mental-fatigue-associated decline in mental task performance with this protocol.

### 4.3. Trait Influence on Fine Motor Task Performance

The findings of our study suggest those who were normally mentally and physically fatigued reported improvement in fine motor task performance with both their dominant and non-dominant hands after consuming caffeine. While caffeine has been known to decrease fine motor task performance [41,42,43], it seems that those who normally report feeling physically or mentally fatigued report acute motor dexterity benefits of consuming caffeine. We are unaware of literature that reports a group of individuals who report improvements in motor task performance after caffeine consumption. However, from the results of our study, we may argue that there is a subset of the population that may respond counter to expectations with caffeine consumption as it relates to performance on fine motor tasks.

### 4.4. Implications

This study deepens our understanding of the influence of long-standing predisposition of mental and physical energy and fatigue on acute responses to caffeine consumption. Our results found that trait mental and physical energy and fatigue may explain many of the interpersonal differences in responses to caffeine consumption. These results provide us with an inexpensive measure to control for inter-individual effects of caffeine and help us identify hyper- and non-responders, or, in the case of fine motor task performance, a group that has the opposite response to caffeine consumption than what has been previously reported in the literature. While most caffeine researchers account for prior night’s sleep [16,17,29] in their studies, we find that researchers should also account for an individual’s trait mental and physical energy and fatigue, as this predisposition may play a role in how individuals respond acutely to caffeine. Additionally, while some researchers utilize random group assignments when performing caffeine research [44,45], our findings suggest that these random group assignments may not control for the inter-individual differences in caffeine response. We recommend the use of a cross-over design where all participants receive all interventions, as it may be best at reducing the risk that individuals who were on one end of one of a trait spectrum are not accidentally assigned to the same group. Therefore, it is the suggestion of the authors that nutrition researchers who are interested in understanding the acute effects of caffeine (1) measure trait mental and physical energy and fatigue in their participants, (2) control for trait mental and physical energy and fatigue in their analyses, and (3) avoid random group assignment when possible and utilize a cross-over design.

### 4.5. Limitations

Like all studies, this study has several aspects that may limit the generalizability of the findings. First, recruitment was limited to average or lower than average consumers of caffeine (<200 mg/day), fruits, vegetables, and other foods rich in polyphenols and POMS vigor scores of <13. Second, although our models reported adequate power, we had a relatively small number of participants and we also did not control for the timing and composition of the meals preceding testing. Third, although we did not find any significant effect of trait moods on pre–post caffeine differences, we did not obtain saliva samples between completion of beverage consumption, the second, and the third mental energy test battery. Therefore, it is unclear if caffeine’s bioavailability between mental energy test batteries varied based on trait. Another limitation of our study is the poor psychometric properties of the POMS survey. While the POMS in healthy populations has been reported to have a Cronbach’s α < 0.90 [31], the α’s for anxiety, depression, anger, and confusion in our study were surprisingly low. Another potential limitation of our study was that we only allowed participants to practice cognitive tasks once prior to administering the protocol, which may have led to learning effects during the study. To mitigate the risk of learning effects, we initially compared all measures by testing days, regardless of the intervention. No significant differences in performance between days were identified. Another possible limitation may have been the carryover effect of one of the beverages on subsequent testing sessions. To mitigate this risk, consistent with prior literature [16,17,26,27,29], we randomized the order of the allocation and scheduled participants a minimum of 48 h after completion of the prior session to allow for a washout period. The average time between treatments was 7.4 ± 2.8 days (i.e., 13.7 ± 3.1 days between the two treatments analyzed in this study). Additionally, some participants may have been more sensitive to the taste of caffeine, thus being able to identify when they were given the placebo versus when they were given the caffeinated beverage.

## 5. Conclusions

The objective of this study was to determine the influence of trait to mental and physical energy and fatigue on state moods and cognitive and fine-motor task performance when participants consumed caffeine compared to days they consumed a placebo. Our results found that trait mental and physical energy and fatigue modified the effects of caffeine on all three parameters and in the instance of fine motor skill had the opposite of the intended effect on participants who reported being high trait mental and physical fatigue. This analysis helps us better identify hyper- and non-responders to caffeine without performing genetic testing. Our results suggest that nutrition researchers should consider trait mental and physical energy and fatigue when conducting studies on acute responses of caffeine on mood, cognitive and fine-motor tasks. Future research should compare these trends on longer time frames (e.g., chronic caffeine consumption).

## Figures and Tables

**Figure 1 nutrients-13-00412-f001:**
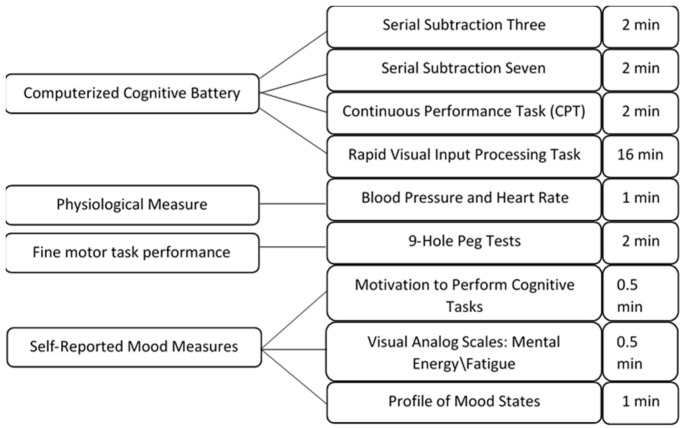
Test battery sequence.

**Figure 2 nutrients-13-00412-f002:**
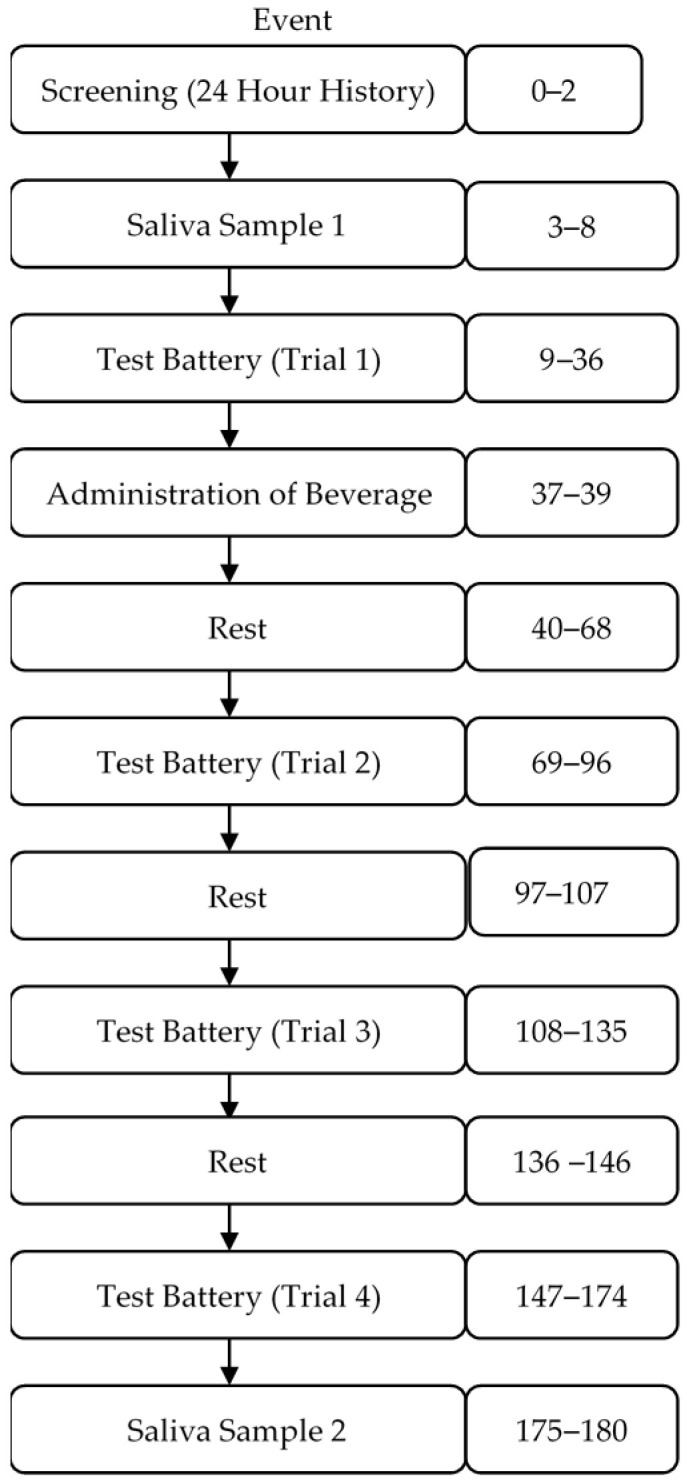
Testing day schedule.

**Figure 3 nutrients-13-00412-f003:**
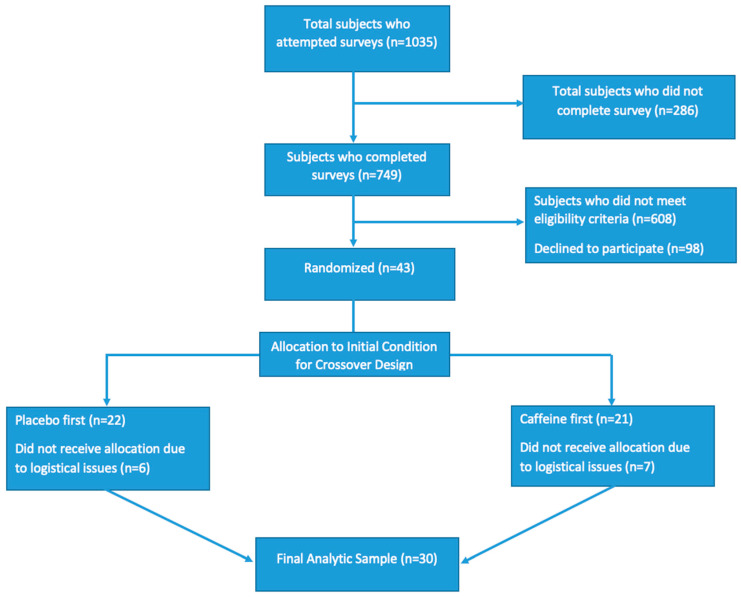
Consolidating Standards of Report Trials (CONSORT) Flow Diagram.

**Figure 4 nutrients-13-00412-f004:**
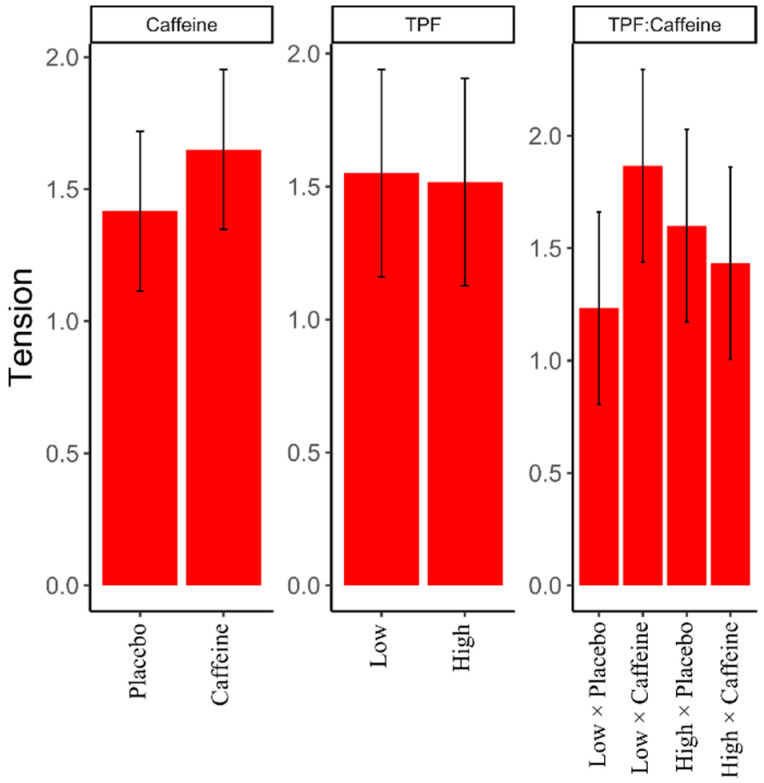
Profile of Mood Survey (POMS) Tension score changes. TPF = Trait Physical Fatigue. Data are presented as means ± standard deviations.

**Figure 5 nutrients-13-00412-f005:**
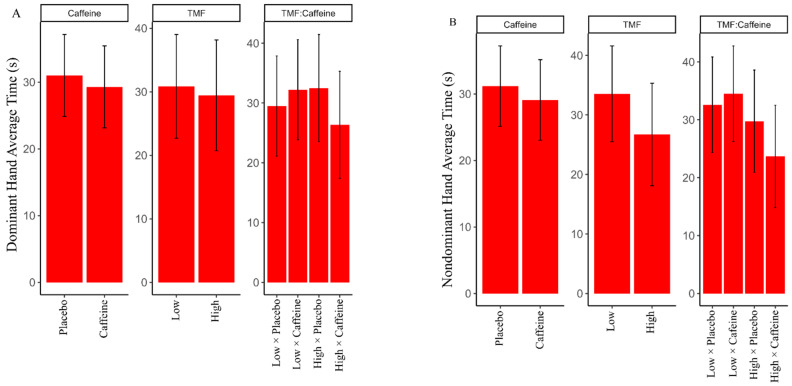
Trait and caffeine influence on nine-hole peg test times for both (**A**) dominant hand and (**B**) nondominant hand average times. TMF = Trait Mental Fatigue.

**Table 1 nutrients-13-00412-t001:** Participant Characteristics.

Sex (Males/Females)	13/17
Age (years)	21.8 ± 4.4
Height (cm)	169.6 ± 12.4
Weight (kg)	67.6 ± 11.0
Body Mass Index (kg/m^2^)	23.5 ± 2.5
**Race**	
White	21
Asian	4
Black	4
More than one race	1
Trait Physical Energy	6.5 ± 2.1
Trait Physical Fatigue	3.5 ± 2.0
Trait Mental Energy	5.7 ± 1.7
Trait Mental Fatigue	3.8 ± 1.9
Amount of sleep on a typical night in the past month (hrs)	7.6 ± 0.8
**Consumption of High-Flavanol Foods or Beverages during the Past Month**	
Caffeine drinks (servings) per week	4.2 ± 3.8
Caffeine drinks (mg caffeine) per day	60 ± 54.3
Cocoa (servings) per month	0.7 ± 1.3
Fruits (servings) per month	12.3 ± 12.4
Vegetables (servings) per month	25.1 ± 14.5

Data are reported as means ± standard deviations.

**Table 2 nutrients-13-00412-t002:** Trait and caffeine influence on moods, cognitive tasks and fine motor tasks (significant relationships only).

Factor	Measure	Beta	2.5%	97.5%	t stat	*p* Value
TPE	Vigor	−3.234	−5.686	−0.783	−2.586	0.014
Caffeine	Vigor	1.300	0.211	2.389	2.341	0.022
Caffeine	Vigor	1.437	0.396	2.479	2.706	0.008
TMF × Caffeine	Vigor	−1.866	−3.390	−0.342	−2.400	0.019
Caffeine	Tension	0.633	0.265	1.001	3.372	0.001
TPF × Caffeine	Tension	−0.800	−1.321	−0.279	−3.012	0.003
Caffeine	Anger	0.400	0.013	0.787	2.028	0.046
Caffeine	Anger	0.364	0.045	0.683	2.234	0.028
TMF × Caffeine	Depression	−0.339	−0.632	−0.047	−2.276	0.025
TPE	Confusion	0.830	0.033	1.628	2.040	0.047
TPE	Motivation	−1.469	−2.827	−0.111	−2.121	0.040
Caffeine	Motivation	0.812	0.094	1.531	2.215	0.029
TMF × Caffeine	Motivation	−1.312	−2.365	−0.260	−2.444	0.017
TPE	Physical Fatigue	−4.926	−8.137	−1.715	−3.007	0.004
TME × Caffeine	Physical Fatigue	5.369	1.502	9.237	2.721	0.008
TPE	Mental Fatigue	−4.483	−7.678	−1.289	−2.751	0.008
TME × Caffeine	Mental Fatigue	5.199	1.306	9.092	2.617	0.010
TPF × Caffeine	Sub3total	4.400	0.796	8.004	2.393	0.019
TME	Sub3total	9.222	2.045	16.398	2.519	0.017
TME × Caffeine	Sub3total	−6.244	−10.242	−2.247	−3.062	0.003
Caffeine	Sub3total	−2.875	−5.290	−0.46	−2.334	0.022
TMF × Caffeine	Sub3total	5.661	2.126	9.196	3.139	0.002
Caffeine	Sub7total	2.632	0.518	4.746	2.440	0.017
TME	Sub7total	6.858	1.106	12.61	2.337	0.025
Caffeine	Sub7total	2.886	0.933	4.840	2.896	0.005
TMF × Caffeine	Sub7total	4.964	1.751	8.178	3.028	0.003
TPF × Caffeine	DH	−9.433	−15.295	−3.572	−3.154	0.002
TMF × Caffeine	DH	−8.862	−14.777	−2.947	−2.936	0.004
TPF × Caffeine	NDH	−8.033	−13.744	−2.322	−2.757	0.007
TMF × Caffeine	NDH	−7.978	−13.706	−2.250	−2.730	0.008

TPE = Trait Physical Energy, TPF = Trait Physical Fatigue, TME = Trait Mental Energy, TMF = Trait Mental Fatigue, DH = Dominant Hand, NDH = Non-dominant hand.

## Data Availability

The dataset supporting the conclusions of this article is available in the Mendley repository. The DOI number for this dataset is 10.17632/3s8sr9zth9.1.

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
