# Peer review of "Trait Energy and Fatigue Modify the Effects of Caffeine on Mood, Cognitive and Fine-Motor Task Performance: A Post-Hoc Study"

_nutrients, 2021, doi:10.3390/nu13020412_

Round 1
Reviewer 1 Report
Dear Authors,
my suggestions and questions are:
- in the introduction, I would add a few words about caffeine, theine or mateine, and about the way caffeine works in coffee and energy drinks, i.e. about its release;
- was the numerical sample too small? only 30 people. Maybe if you continue the research you write about, you will expand the control group in addition to the research period;
- 2.3 point 4). the respondents used the dominant and then the recessive hand 2 times? average of 4 tests?
- table 2 - can these results be presented differently? for now it's tedious reading 3 pages of numbers;
Generally a very correct manuscript.
Reviewer 2 Report
This is an interesting post-hoc analysis of previous research which adds a useful contribution to the subject area. However, it requires revision in some areas and suggestions for improvement are detailed below.
Introduction
Line 51 ‘substantial evidence’ is referred to here without supporting references. Please include.
Methods
If participants were only required to practise the cognitive tasks once prior to their testing visits it is likely that learning effects would still be present. This should be addressed.
Measuring only total attempts for the serial subtractions tasks provides limited scope for interpretation since number of attempts does not indicate improvements in performance (participants could attempt more but get more wrong, or be disengaged with the task entirely and still score highly). This is particularly relevant in a study looking at fatigue. This should be addressed in the discussion, so the reader does not mistakenly draw inaccurate conclusions.
It would be useful to include in the description of the fine motor control task, what this task was measured in.
Table 1 – It should be made clearer within the table at a glance what period of time the consumption levels of each food and beverage are measured within (for example, caffeine is given within the week, however others are given within the month).
The participants within this study appear to be low consumers of caffeine (< 1 caffeinated beverage per day in most cases). It would be useful to include their average daily levels of caffeine consumption in mg in addition (or, as opposed) to ‘servings’ in table 1.
Discussion
Line 301 sentence does not make sense – consider including ‘as being’.
Line 324 remove ‘are’.
Please see above comments regarding practice effects.
Please see above comments regarding serial subtractions and measuring total attempts.
